# Complete Mitochondrial Genomes of Two Water Mite Species in the Family Sperchontidae (Acari: Hydrachnidiae): Characterization and Phylogenetic Implications

**DOI:** 10.3390/genes16101236

**Published:** 2025-10-19

**Authors:** Xu Zhang, Xingru Nie, Xuhang Xia, Jiahui Song, Qingyu Wen, Ke Sun

**Affiliations:** School of Life Sciences, Huaibei Normal University, Huaibei 235000, China; 15256521262@163.com (X.N.); 20231504036@chnu.edu.cn (X.X.); song18726825878@outlook.com (J.S.); wen56700@outlook.com (Q.W.); 20231501068@chnu.edu.cn (K.S.)

**Keywords:** *Sperchon*, Trombidiformes, Acari, mitogenomics, gene order

## Abstract

**Background:** The family Sperchontidae Thor, 1900 is proposed as a transitional group between the “lower” and “higher” water mites (Subcohort Hydrachnidiae), and is important for understanding the evolutionary history of Hydrachnidiae. However, mitogenomic data are lacking. **Methods:** The first complete mitogenomes of Sperchontidae were sequenced from two species, *Sperchon plumifer* and *Sperchon* sp. Structural features were analyzed, gene rearrangements were compared with five published water mite mitogenomes, and phylogenetic relationships among 31 species within the order Trombidiformes were reconstructed. **Results:** Both mitogenomes contained the typical 37 genes and exhibited a strong A+T bias (73.1–73.6%), positive AT-skew, and negative GC-skew. Protein-coding genes (PCGs) were generally initiated with ATN/TTG codons and terminated with TAA/TAG or incomplete T–, with codon usage biased toward T/U-ending codons; all PCGs were under purifying selection (Ka/Ks < 1). Most tRNAs lacked canonical cloverleaf structures due to D- or T-arm loss. Gene rearrangements occurred in all examined water mite mitogenomes, with intrageneric rearrangements restricted to tRNAs in Hygrobatidae and Unionicolidae but involving both tRNAs and PCGs in Sperchontidae. Phylogenetic analyses using ML and BI (13 PCGs + 2 rRNAs) strongly supported a close relationship between Hydrachnidiae and Trombidiae (BS = 100%, PP = 1.00) and confirmed the three supercohorts in Trombidiformes (Eleutherengonides, Anystides, Eupodides), though relationships among them remained unresolved. **Conclusions:** This study reports the first two complete mitogenomes of Sperchontidae, providing preliminary insights into gene rearrangement patterns in water mites. The phylogenetic analyses based on mitochondrial genomes provide additional support for the consistency with traditional morphology at lower taxonomic levels, such as within genera and families, whereas relationships among supercohort-level taxa remain unstable and require additional data for further clarification.

## 1. Introduction

Water mites (Subcohort Hydrachnidiae) are a diverse group of mites inhabiting aquatic environments, and occurring in all types of freshwater bodies, including springs, streams, lakes, and ponds. They are distributed worldwide, except for Antarctica, and represent one of the most abundant and diverse groups of arthropods in freshwater ecosystems [1]. To date, more than 7500 species belonging to 56 families have been described [2].

Within Hydrachnidiae, the family Sperchontidae Thor, 1900 is of particular interest due to its putative evolutionary position. A transitional group between “lower” and “higher” water mites is proposed because its larvae combine features of both groups: they are adapted for underwater host-seeking like higher mites, yet retain smaller coxae and variable dorsal plates reminiscent of lower mites [3,4,5]. Thus, clarifying its phylogeny is essential for understanding the evolutionary history of Hydrachnidiae. However, studies on Sperchontidae have remained largely focused on morphology [6,7,8,9]. Although more than 200 species have been reported worldwide, their phylogenetic relationships, particularly at the molecular level, are still poorly understood. Available molecular data are scarce and limited to a few COI sequences, while genomic information remains absent.

With the rapid development of molecular biology, mitochondrial genomes have attracted increasing attention. Compared with single-gene markers, the mitogenome provides a rich source of phylogenetically informative features, including 13 protein-coding genes, 2 ribosomal RNAs, and 22 transfer RNAs, as well as information on gene order and RNA secondary structures. As a relatively small and maternally inherited genome, its gene content is highly conserved, yet its sequences and structures evolve quickly. These properties make the mitogenome a powerful tool in molecular systematics and suitable for resolving relationships at various taxonomic levels. Despite these advantages, the application of mitogenomic data in water mite research has been very limited. To date, only five complete mitogenomes have been reported: *Hygrobates longiporus* Thor, 1898, *Hygrobates taniguchii* Imamura, 1954, *Hygrobates turcicus* Pešić, Esen & Dabert, 2017, *Unionicola foili* Edwards & Vidrine, 1994 and *Unionicola parkeri* Vidrine, 1987 [10,11,12,13]. Critically, no mitogenome has yet been published for any member of Sperchontidae. This absence has left a major gap in the knowledge of this transitional family and has hindered progress in the phylogenetic study of water mites.

The present study reports the first complete mitogenomes of Sperchontidae, based on two species: *Sperchon plumifer* and *Sperchon* sp. Structural features of their mitogenomes are described, gene rearrangements are examined in comparison with other water mite mitogenomes, and phylogenetic relationships among 31 Trombidiformes species are reconstructed. Given the scarcity of genomic data for this family, these results provide the first mitogenomic data for Sperchontidae, and contribute to understanding the evolutionary history of Hydrachnidiae

## 2. Materials and Methods

### 2.1. Sample Collection and Identification

Specimens of *S. plumifer* were collected from an unnamed stream in Fuxi Village, Anhui Province, China (30°04′16″ N, 118°09′26″ E) on 26 May 2017. Specimens of *Sperchon* sp. were collected on 16 February 2021 from an unnamed stream in Yinhui Town, Chizhou City, Anhui Province, China (30°25′16″ N, 117°22′41″ E). All specimens were identified based on morphology by the first author. Voucher specimens were deposited in the College of Life Sciences, Huaibei Normal University, Huaibei, China. Samples for molecular analysis were preserved in 100% ethanol and stored at −20 °C until DNA extraction.

### 2.2. Genomic DNA Extraction and PCR Amplification

Total genomic DNA was extracted from a single specimen of each species using the DNeasy Blood & Tissue Kit (Qiagen, Hilden, Germany) following the manufacturer’s instructions. Before extraction, each specimen was rinsed several times with sterile deionized water.

The complete mitogenome of each species was amplified in two overlapping long PCR fragments. First, two conserved mitochondrial gene fragments—cytochrome c oxidase subunit I (*cox1*) and cytochrome b (*cob*)—were amplified with universal primers (Table 1). PCR reactions were performed in 25 μL volumes containing 12.5 μL of 2× Premix Taq™ (Takara, Dalian, China), 0.4 μM of each primer, and 2 μL of genomic DNA. The thermal program included 95 °C for 5 min, followed by 30 cycles of 95 °C for 30 s, annealing at 46–51 °C for 30 s, and 72 °C for 45 s, with a final extension at 72 °C for 10 min. Based on the *cox1* and *cob* sequences, species-specific long PCR primers were designed (Table 1). Two overlapping fragments were then amplified: a long fragment (~9.8 kb, *cox1* to *cob*) and a short fragment (~4.8 kb, *cob* to *cox1*). Amplifications used PrimeSTAR GXL DNA Polymerase (Takara). Each 25 μL reaction contained 5 μL of 5× PrimeSTAR GXL Buffer, 2 μL of dNTPs (2.5 mM each), 0.4 μM of each primer, 0.5 μL of polymerase, and 2 μL of DNA template. The thermal profile was 98 °C for 10 s, 60 °C for 15 s, and 68 °C for 5–10 min (extension time adjusted to fragment length), repeated for 30 cycles.

The sequencing strategy differed between the two species. For *S. plumifer*, the long PCR products were directly sequenced with the Sanger method using primer walking at General Biol, Chuzhou, China. For *Sperchon* sp., the two amplicons were pooled and sequenced on an Illumina NovaSeq PE150 platform at Genepioneer, Nanjing, China. Paired-end reads were assembled de novo with SPAdes v3.15 to recover the long fragments.

### 2.3. Mitogenome Assembly, Annotation and Analysis

The complete mitogenomes of both species were obtained by merging the two overlapping fragments in Geneious Prime 11.1.5. Initial annotation was carried out with the MITOS web server on the Galaxy platform [15]. Protein-coding genes (PCGs) and ribosomal RNA (rRNA) boundaries were determined by alignment with homologous genes from related species. Transfer RNA (tRNA) genes were detected with MITOS, ARWEN v1.2.3 [16], and tRNAscan-SE 2.0 [17], followed by manual adjustment. Predicted tRNA secondary structures were visualized using VARNA v3-93 [18].

Nucleotide composition and strand asymmetry (AT- and GC-skews) were calculated in Geneious Prime using the formulas AT-skew = (A − T)/(A + T) and GC-skew = (G − C)/(G + C). Relative synonymous codon usage (RSCU) for concatenated PCGs and gene order comparison were conducted in PhyloSuite v1.2.2 [19]. The nonsynonymous (Ka) and synonymous (Ks) substitution rates of the 13 PCGs were calculated with DnaSP 6.0 [20]. The Ka/Ks ratio for each gene was used to evaluate evolutionary rates. Circular genome maps were generated using Proksee (https://proksee.ca/, accessed on 31 August 2025).

### 2.4. Phylogenetic Analysis

To investigate phylogenetic relationships within Trombidiformes, a dataset of 35 complete mitogenomes was built. It included the two new *Sperchon* species and 33 additional mitogenomes from GenBank. The in-group contained 31 Trombidiformes species, and 4 Sarcoptiformes species were used as the out-group, since Sarcoptiformes is the sister group of Trombidiformeses, and together they form the monophyletic Acariformes [21,22]. To make the phylogenetic analysis more reliable, only complete, published mitogenomes were included in the analysis. Taxa from the superfamily Eriophyoidea were excluded, as multiple studies have shown they do not belong to Trombidiformes [23,24,25,26,27]. A full list of taxa and accession numbers is provided in Table 2.

The dataset consisted of 13 PCGs and 2 rRNA genes, extracted using PhyloSuite v1.2.2. Each PCG was aligned with MAFFT v7.450 and refined with MACSE v2.03 for codon accuracy. Ambiguously aligned regions were removed with Gblocks v0.91b. The rRNA genes were aligned with MAFFT and trimmed with trimAl v1.4.1. All alignments were concatenated into a single matrix. ModelFinder in PhyloSuite was used to determine the best partitioning scheme and substitution models. Phylogenetic trees were inferred with Maximum Likelihood (ML) and Bayesian Inference (BI). ML analysis was conducted in IQ-TREE v2.1.2 [45] with 1000 ultrafast bootstrap replicates (BS). BI analysis was carried out in MrBayes v3.2.6 [46]. Two independent runs with four chains (one cold, three heated) were performed for 5 million generations, sampling every 1000 generations. Convergence was assessed by split frequency (<0.01) and effective sample size (ESS > 200) in Tracer v1.7.1. The first 25% of trees were discarded as burn-in, and a 50% majority-rule consensus tree was generated to calculate posterior probabilities (PP).

## 3. Results

### 3.1. Mitogenome Organization and Base Composition

The complete mitochondrial genomes of *S. plumifer* and *Sperchon* sp. were sequenced and assembled into circular, double-stranded DNA molecules (Figure 1). The mitogenome of *S. plumifer* is 14,646 bp in length, while that of *Sperchon* sp. is 14,724 bp. Both mitogenomes contain the typical set of 37 mitochondrial genes: 13 protein-coding genes (PCGs), 22 transfer RNA (tRNA) genes, and 2 ribosomal RNA (rRNA) genes. The gene distribution across the two strands is conserved in both species. A total of 22 genes is encoded on the majority (J) strand, comprising nine PCGs and thirteen tRNAs. Consequently, the remaining 15 genes are located on the minority (N) strand, including the other four PCGs, nine tRNAs, and both rRNA genes (*rrnL* and *rrnS*). The nucleotide composition was highly biased towards A and T, with overall A+T contents of 73.6% in *S. plumifer* and 73.1% in *Sperchon* sp. Both genomes exhibited a positive AT-skew (*S. plumifer*: 0.224; *Sperchon* sp.: 0.260) and a negative GC-skew *(S. plumifer*: −0.183; *Sperchon* sp.: −0.231) on the J-strand, indicating a higher abundance of A over T and C over G. The genomes are compact, with short intergenic spacers and some gene overlaps. In *S. plumifer*, intergenic spacers range from 1 to 50 bp, while gene overlaps span 1 to 9 bp. *Sperchon* sp. possesses intergenic spacers range from 1 to 39 bp and gene overlaps span 1 to 8 bp. Detailed information for each gene junction can be found in Appendix A.

### 3.2. Protein-Coding Genes and Codon Usage

All 13 PCGs in the two *Sperchon* mitogenomes initiated with ATN codons, with the exception of *nad1* and *nad5*, which utilized the alternative start codon TTG in both species. The canonical start codon ATG was used for six genes in *S. plumifer* and seven in *Sperchon* sp., while ATA initiated translation for four and three genes, respectively. The *atp8* gene started with ATC in *S. plumifer* and ATT in *Sperchon* sp. For translation termination, both complete and incomplete stop codons were identified. TAA was the most common stop codon, terminating nine PCGs in *S. plumifer* and eight in *Sperchon* sp. The TAG codon terminated *nad1* gene in *S. plumifer* and two genes (*nad4L*, *nad4*) in *Sperchon* sp. In both species, three genes (*cox2*, *cox3*, *nad5*) ended with an abbreviated stop codon (T-), which likely becomes a functional TAA codon after post-transcriptional polyadenylation (Appendix A).

The analysis of relative synonymous codon usage (RSCU) revealed a strong and consistent bias across all PCGs in both *Sperchon* mitogenomes, a pattern consistent with their high A+T nucleotide composition (Figure 2). A clear codon usage bias was detected, with codons ending in T/U being more frequent at the third position and those ending in A or G being strongly underrepresented. For example, for glycine (Gly), GGU was the most frequently used codon in both species, whereas GGC, GGA, and GGG were rarely used. Similarly, the most frequently used codons for Alanine, Valine, and Threonine were GCU, GUU, and ACU, respectively. This codon usage pattern was nearly identical between *S. plumifer* and *Sperchon* sp., indicating conserved evolutionary pressures shaping their mitogenomic composition.

To evaluate the selective pressures on the PCGs, the ratio of nonsynonymous (Ka) to synonymous (Ks) substitution rates was calculated for each gene between the two species (Figure 3). All Ka/Ks ratios were substantially below 1, ranging from 0.039 (*cox1*) to 0.444 (*atp8*), indicating that all 13 PCGs are evolving under strong purifying selection. The intensity of this selection pressure, however, varied among genes. The PCGs ranked from most to least conserved were: *cox1* < *cox2* < *cob* < *atp6* < *cox3* < *nad1* < *nad3* < *nad2* < *nad4* < *nad4L* < *nad5* < *nad6* < *atp8*. This identifies *cox1* as the most conserved gene, while *atp8* is evolving under the most relaxed purifying selection.

### 3.3. Secondary Structure of tRNAs

The full complement of 22 tRNA genes was identified in both *Sperchon* mitogenomes, with lengths ranging from 49 to 64 bp (average was 56.2) in *S. plumifer* and 50 to 61 bp (55.9) in *Sperchon* sp. (Appendix A). A pervasive loss of the canonical cloverleaf secondary structure was observed in the majority of these tRNAs (Figure 4). Only a minority of tRNAs could be folded into a typical cloverleaf shape. Specifically, in *S. plumifer*, only seven tRNAs (*trnD*, *trnK*, *trnL1*, *trnL2*, *trnM*, *trnN*, *trnR*) retained a complete cloverleaf structure. Similarly, only six tRNAs in *Sperchon* sp. (*trnD*, *trnK*, *trnL1*, *trnL2*, *trnM*, *trnN*) were found to be canonical. The majority of tRNAs exhibited highly truncated structures, primarily due to the loss of either the D-arm or the T-arm: six tRNAs in *S. plumifer* (*trnA*, *trnG*, *trnI*, *trnS1*, *trnS2*, *trnV*) and seven in *Sperchon* sp. (*trnA*, *trnG*, *trnI*, *trnR*, *trnS1*, *trnS2*, *trnV*) lacked a D-arm. Nine tRNAs in both *Sperchon* species (*trnC*, *trnE*, *trnF*, *trnH*, *trnP*, *trnQ*, *trnT*, *trnW*, *trnY*) were missing a T-arm. While the secondary structures of 21 tRNAs were largely conserved between the two species, notable structural differences were observed in *trnR*. For instance, *trnR* possessed a full cloverleaf structure in *S. plumifer* but lacks a D-arm in *Sperchon* sp. Furthermore, some mismatched base pairs, including G-U, U-U, A-G, A-A, and C-A, were identified across the stem regions of many tRNAs.

### 3.4. Mitochondrial Gene Order and Rearrangements

The gene order of the two newly sequenced *Sperchon* mitogenomes, along with five previously published water mite mitogenomes, was compared to the putative ancestral arthropod [47]. All seven water mites mitogenomes showed extensive gene rearrangements relative to this ancestral state (Figure 5). Among them, six distinct gene orders were identified: only *H. longiporus* and *H. taniguchii* shared an identical arrangement, while each of the other five species possessed a unique gene order. Gene rearrangements among congeneric species in other genera are restricted to tRNA genes. In contrast, rearrangements within the genus *Sperchon* (Sperchontidae) involve not only tRNA genes but also PCGs. Despite this high variability, three gene clusters were perfectly conserved across all seven water mites: *cox3*-*trnE*, *nad6*-*cytb*-*trnS2*-*rrnS*-*trnP*-*nad1*- *trnL2*-*rrnL* and *trnQ*-*trnM*-*nad2*.

### 3.5. Phylogenetic Analysis of Trombidiformes

Phylogenetic trees were reconstructed based on the concatenated dataset of 13 PCGs and two rRNAs. The topologies obtained from Maximum Likelihood (ML) and Bayesian Inference (BI) analyses were largely congruent (Figure 6). Both analyses strongly supported the monophyly of Hydrachnidiae (water mites), with all seven species forming a single, well-supported clade (ML bootstrap support [BS] = 100%; Bayesian posterior probability [PP] = 1.00). In addition, Hydrachnidiae clustered with four species of Trombiculoidea (subcohort Trombidiae), forming a strongly supported clade (BS = 99%; PP = 1.00). Furthermore, three major Trombidiformes lineages were consistently resolved with strong support, in agreement with traditional classification: (1) Eleutherengonides (Tetranychoidea + Cheyletoidea + Pyemotoidea) (BS = 100%; PP = 1.00), (2) Anystides (Hygrobatoidea + Lebertioidea + Trombiculoidea) (BS = 100%; PP = 1.00), and (3) Eupodides (Bdelloidea + Eupodoidea + Tydeoidea) (BS = 91%; PP = 1.00). However, the relationships among these three clades remained unstable and represented the main source of conflict between the two inference methods. The BI analysis supported the topology [Eupodides + (Anystides + Eleutherengonides)], but with low posterior probability (PP = 0.598). In contrast, the ML analysis favored [Eleutherengonides + (Anystides + Eupodides)], also with low bootstrap support (BS = 48%).

## 4. Discussion

The mitogenomes of *S. plumifer* and *Sperchon* sp. sequenced in this study exhibit typical features of water mites, including a compact size, the standard set of 37 genes, and a strong A+T bias. A key difference lies in the number of control regions (CRs). *S. plumifer* has a single CR, whereas *Sperchon* sp. has two. This pattern is consistent with the variation observed in other water mites. For example, *U. parkeri* has one CR [13], while *H. turcicus* and *U. foili* each possess two [10,12]. In contrast, annotations of *H. longiporus* and *H. taniguchii* report a complete absence of the CR [11]. Since the CR is essential for initiating replication and transcription of the mitogenome, its absence is unlikely to be genuine. Thus, the reported lack of CRs in some species is probably due to annotation errors, highlighting the need for a careful re-examination of these genomes. The evolutionary rate analysis across 13 PCGs indicated strong purifying selection (Ka/Ks < 1), indicating functional constraint on all mitochondrial proteins. As expected, *cox1* was the most conserved gene, reinforcing its suitability as the universal DNA barcode for species identification [48,49,50]. Conversely, *atp8* was identified as the most rapidly evolving gene. This high evolutionary rate is common across many arthropod lineages and is often attributed to its small size and its role as a structural, rather than catalytic, component of the ATP synthase complex, suggesting lower functional constraints [51,52,53].

Typically, metazoan tRNAs fold into a canonical cloverleaf structure with four arms: the amino acid acceptor arm (AA-arm), the D-arm, the anticodon arm (AC-arm), and the T-arm. However, the loss of the D-arm or T-arm is a widespread phenomenon in Acari [24,33,40,54,55]. As a consequence, automated annotation tools such as MITOS and tRNAscan-SE often fail to detect many tRNAs, while ARWEN frequently requires extensive manual curation due to low-confidence predictions. In extreme cases, tRNAs with severely truncated structures can only be identified through manual inference based on conserved anticodon sequences. Despite these methodological challenges, all 22 tRNAs were successfully annotated in the two *Sperchon* species and confirmed multiple instances of arm loss (see Results for details)—a pattern consistent with previously reported features in Acari. Notably, whereas a previous investigation of two species in the same genus *Unionicola* reported an almost identical pattern of arm loss across all 22 tRNAs [12,13], our results revealed subtle differences in the genus *Sperchon*: the overall secondary structures were largely conserved, except for variations in *trnR*. Furthermore, nucleotide mismatches within tRNA stems, another common feature in Acari [56], were also observed in this study. It is hypothesized that these mismatches are corrected by post-transcriptional RNA editing mechanisms to maintain proper tRNA function [57,58]. The mitochondrial tRNAs in Acari often undergo significant sequence reduction, making the accurate annotation of these non-canonical structures difficult, especially without sufficient data from closely related species for comparison. Therefore, sequencing additional mitogenomes from a broader range of water mites is crucial. Such efforts will not only improve the accuracy of mitogenome annotation but also provide a more robust foundation for advancing the field of molecular systematics in water mites.

Ernsting et al. and Edwards et al. were the first to report complete mitochondrial genomes of two water mite species, *U. foili* and *U. parkeri*, highlighting the presence of extensive gene rearrangements—even between congeneric species [12,13]. Following these findings, our comparative analysis of seven water mite mitogenomes representing three genera shows that all species have undergone dramatic rearrangements compared to the ancestral arthropod gene order. Remarkably, six distinct rearrangement patterns were identified among these seven species, with only *H. longiporus* and *H. taniguchii* sharing the same pattern. These findings suggest that intrageneric rearrangements may occur in water mites, and that gene order conservation appears to be uncommon even among closely related species. However, given that this conclusion is based on a limited dataset of seven mitogenomes from three genera, these patterns should be considered preliminary and require confirmation with broader taxon sampling. Notably, the patterns of rearrangement varied among different families. In Hygrobatidae and Unionicolidae, intrageneric rearrangements were limited to tRNA genes, while protein-coding genes (PCGs) remained stable. In contrast, rearrangements in Sperchontidae exhibited rearrangements involving both tRNAs and PCGs. This finding indicates a clear difference in the mode of mitochondrial genome evolution within this family compared with Hygrobatidae and Unionicolidae. Although it is tempting to speculate that this distinct pattern of genomic plasticity may be related to the proposed ‘transitional’ position of Sperchontidae, the available data do not provide direct evidence for such a connection. This observation should therefore be regarded as an important empirical result that merits further investigation as additional mitogenomes from related families become available. Despite the high degree of gene rearrangement observed, three gene clusters were found to be highly conserved across all seven water mite species: *cox3-trnE, nad6-cytb-trnS2-rrnS-trnP-nad1-trnL2-rrnL and trnQ-trnM-nad2.* These conserved clusters closely resemble the putative ancestral gene order of Trombidiformes proposed by Fang et al. [27]. Whether these conserved clusters represent synapomorphic features for the water mite lineage is a key question for future research.

In this study, phylogenetic trees were reconstructed using both Maximum Likelihood (ML) and Bayesian Inference (BI) methods. The results reveal a dual aspect of mitochondrial genome utility in Trombidiformes systematics. On one hand, the results showed that mitochondrial genome data provide strong resolution below the supercohort level, with high support at the superfamily, family, and genus levels. Within Hydrachnidiae, all seven species formed a strongly supported monophyletic group. These findings demonstrate that despite extensive gene rearrangements, mitochondrial genomes still retain reliable phylogenetic signals and remain a valuable tool for resolving relationships among water mites at lower taxonomic levels. On the other hand, our analyses also help clarify long-standing debates about the position of Hydrachnidiae within Trombidiformes. Based on biology, such as larval parasitism and nymphal, adult predation, Hydrachnidiae were thought to be closely related to Erythraeidae, Trombidiae, and Stygothrombiae. Dabert et al. [59] also supported this view based on an analysis of 18S rRNA, 28S rRNA, and COI fragments. However, Thia et al. [28] proposed a different hypothesis, suggesting that *Demodex* is the sister group of Hygrobatoidea. The present results rejected this and supported Hydrachnidiae as the sister group of Trombidiae with strong support. This provides solid molecular evidence for the traditional view. However, conflicts were observed between the topologies of the ML and BI trees in resolving high-level relationships among the supercohort level. The BI tree supported Eupodides + (Anystides + Eleutherengonides), whereas the ML tree supported Eleutherengonides + (Anystides + Eupodides). These conflicting topologies, coupled with weak statistical support at these deep nodes, exemplify a well-documented challenge in molecular systematics: the limited resolving power of mitochondrial data for ancient divergences [60,61]. Such difficulties are often attributed to confounding factors inherent to mitogenomes, including substitution saturation, compositional bias, and rate heterogeneity across lineages. Our results suggest that these factors likely affect the deep phylogenetic inference within Trombidiformes, leading to instability in tree topology. Therefore, to confidently resolve these fundamental relationships, future phylogenetic studies will likely require the integration of more slowly evolving nuclear genes and an expanded taxon sampling to build a more robust phylogenetic framework for Trombidiformes.

## 5. Conclusions

This study reports the first complete mitochondrial genomes of *S. plumifer* and *Sperchon* sp., providing essential genomic data for the family Sperchontidae. Both genomes exhibit typical mitochondrial features of water mites, including strong A+T bias, gene rearrangements, and tRNA structural variation. Phylogenetic analyses based on mitochondrial data support traditional morphological classifications and demonstrate that mitogenomes retain reliable phylogenetic signals at lower taxonomic levels, such as within genera and families. However, relationships at higher levels, particularly among the supercohorts of Trombidiformes, remain unresolved, indicating the limitations of mitochondrial data alone. Future studies should expand taxon sampling and integrate nuclear genomic evidence to achieve a more robust framework for understanding Trombidiformes’ evolution.

## Figures and Tables

**Figure 1 genes-16-01236-f001:**
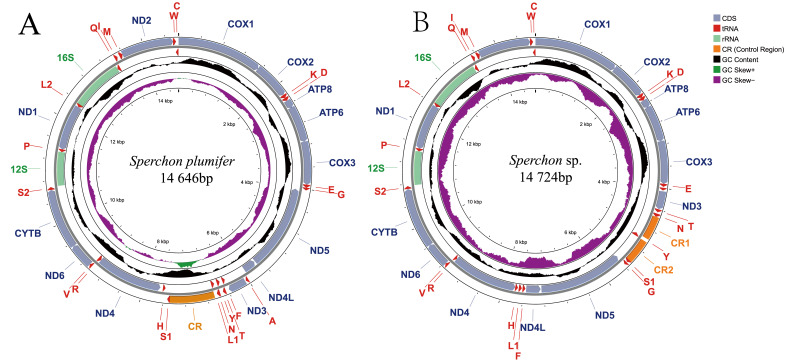
Circular maps of the mitochondrial genomes of (**A**) *Sperchon plumifer* and (**B**) *Sperchon* sp. Genes are color-coded by function as shown in the legend. Genes located on the outer circle are encoded on the J-strand (majority strand), while genes on the inner circle are encoded on the N-strand (minority strand). The innermost ring displays the GC content (black plot) and GC-skew (purple/green plot), where GC-skew = (G − C)/(G + C). The green area in the GC-skew plot indicates positive values, and the purple area indicates negative values.

**Figure 2 genes-16-01236-f002:**
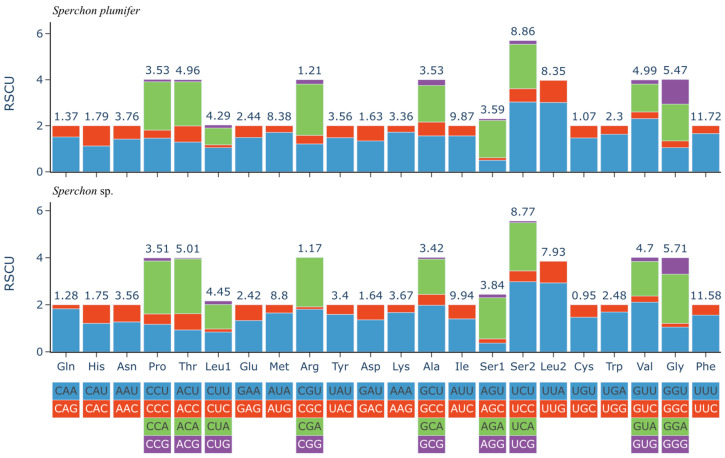
Relative synonymous codon usage (RSCU) in the mitochondrial genomes of *Sperchon plumifer* and *Sperchon* sp. The *x*-axis lists the amino acids, and the *y*-axis represents the RSCU value. The stacked bars for each amino acid show the RSCU values of its synonymous codons. The color of each segment corresponds to a specific codon, as indicated in the key below the *x*-axis. The total number of codons for each amino acid is shown above the respective bar.

**Figure 3 genes-16-01236-f003:**
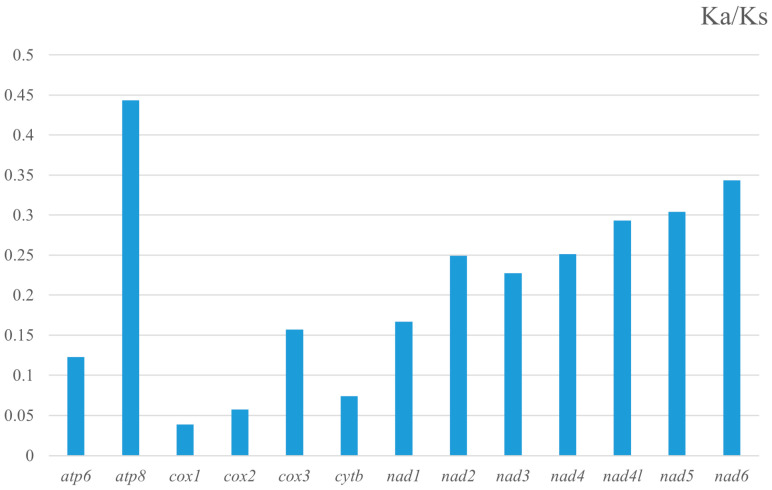
The ratio of non-synonymous (Ka) to synonymous (Ks) substitution rates for the 13 mitochondrial protein-coding genes between *Sperchon plumifer* and *Sperchon* sp. The *x*-axis lists the individual genes, and the *y*-axis indicates the Ka/Ks value.

**Figure 4 genes-16-01236-f004:**
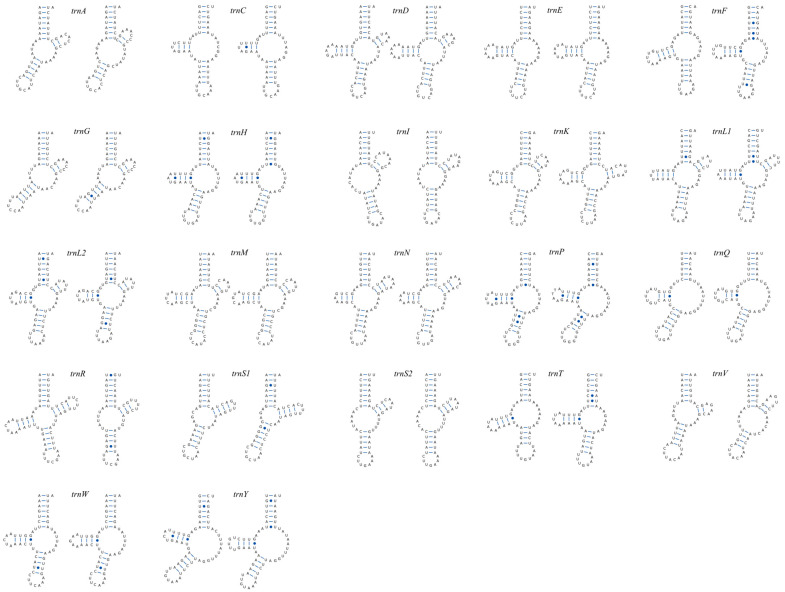
Predicted secondary structures of the 22 transfer RNA (tRNA) genes identified in the mitochondrial genomes of *Sperchon plumifer* and *Sperchon* sp. For each pair, the structure from *S. plumifer* is on the left, and the structure from *Sperchon* sp. is on the right.

**Figure 5 genes-16-01236-f005:**
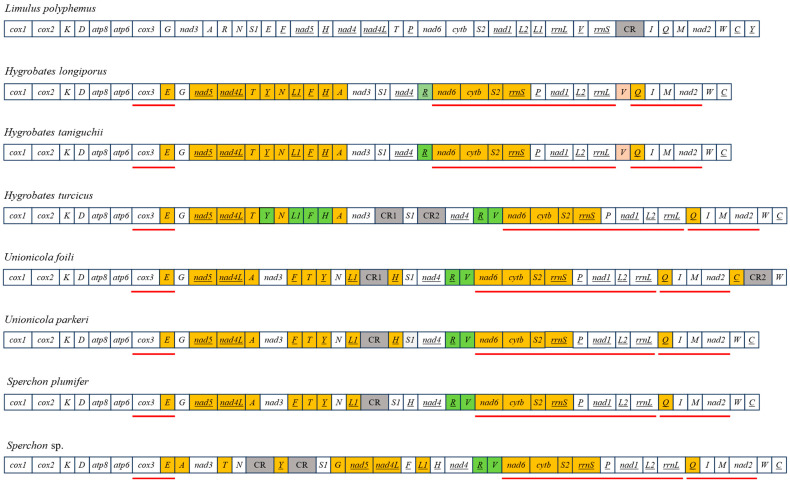
Linearized mitochondrial gene arrangements of seven water mite species, compared with the putative ancestral arrangement of *Limulus polyphemus* (top). Genes encoded on the N-strand are underlined. The color-coding indicates the type of rearrangement event: orange for translocation only, pink for inversion only, and green for genes that have undergone both translocation and inversion. Control regions (CR) are colored gray. The red lines below the gene blocks highlight conserved clusters shared among the analyzed water mite species.

**Figure 6 genes-16-01236-f006:**
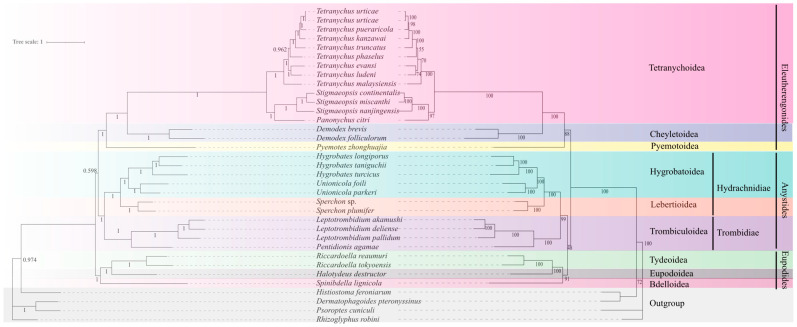
Phylogenetic relationships of Trombidiformes inferred from the concatenated nucleotide sequences of 13 protein-coding genes (PCGs) and 2 rRNA genes. The tree on the left was inferred using Bayesian Inference (BI) and the one on the right using Maximum Likelihood (ML). Nodal support is indicated by Bayesian posterior probabilities (PP) for the BI tree and bootstrap values (BS) for the ML tree.

**Table 1 genes-16-01236-t001:** PCR primers used for the amplification mitochondrial genomes of *Sperchon plumifer* and *Sperchon* sp.

	Primer Name	Amplified Fragment	Sequence (5′–3′)	Reference
	LCO1490	*cox* *1*	GGTCAACAAATCATAAAGATATTGG	[14]
	HCO2198		TAAACTTCAGGGTGACCAAAAAATCA	
	COB-F	*cob*	TTTTGAGGTGCAACAGTAATTAC	[12]
	COB-R		AAATATCATTCAGGTTGAATATG	
*S. plumifer*	pluIBF	*cox1*—*cob*	CAGGAACAGGATGAACAGTTTACCC	This study
	pluIBR		AGTTTATCTAAGTCTCTTTTTATCCCAGTT	
	pluBIF	*cob*—*cox1*	CCCCAATTTATTAGGAGACCCAGAA	
	pluBIR		CCTAAGATAGATGAAACACCCGCCA	
*Sperchon* sp.	spIBF	*cox1*—*cob*	ATGAACAGTCTATCCACCTTTATCT	
	spIBR		TAAGATGAATGGTAATAGAAAATGAAAT	
	spBIF	*cob*—*cox1*	TGTCCAATGATTATGAGGAGGGTTC	
	spBIR		ACGAAAGCGTGTGCTGTAACAATGG	

**Table 2 genes-16-01236-t002:** Information of 35 mitochondrial genomes used in this study.

Order	Supercohort	Superfamily	Family	Species	GenBank Accession Number	Reference
Trombidiformes	Eupodides	Bdelloidea	Bdellidae	*Spinibdella lignicola*	NC067576	[27]
		Eupodoidea	Penthaleidae	*Halotydeus destructor*	NC063625	[28]
		Tydeoidea	Ereynetidae	*Riccardoella tokyoensis*	LC601992	[29]
				*R. reaumuri*	LC601993	[29]
	Anystides	Trombiculoidea	Trombiculidae	*Leptotrombidium pallidum*	NC007177	[30]
				*L. deliense*	NC007600	[31]
				*L. akamushi*	NC007601	[31]
				*Pentidionis agamae*	NC086514	[32]
		Hygrobatoidea	Hygrobatidae	*H. longiporus*	LC552026	[11]
				*H. taniguchii*	LC552027	[11]
				*H. turcicus*	NC068260	[10]
			Unionicolidae	*Unionicola foili*	NC011036	[12]
				*U. parkeri*	NC014683	[13]
		Lebertioidea	Sperchontidae	*Sperchon plumifer*	NC039813	This study
				*Sperchon* sp.	PX252369	This study
	Eleutherengonides	Tetranychoidea	Tetranychidae	*Panonychus citri*	HM189212	[33]
				*Tetranychus urticae*	KJ729023	[34]
				*T. evansi*	MN417333	[35]
				*Stigmaeopsis miscanthi*	MZ726369	[36]
				*T. urticae*	NC010526	[37]
				*T. kanzawai*	NC024676	[34]
				*T. ludeni*	NC024677	[34]
				*T. malaysiensis*	NC024678	[34]
				*T. phaselus*	NC024679	[34]
				*T. pueraricola*	NC024680	[34]
				*T. truncatus*	NC024874	[38]
				*S. nanjingensis*	NC082149	[39]
				*S. continentalis*	NC082150	[39]
		Cheyletoidea	Demodicidae	*Demodex brevis*	KM114225	[40]
				*D. folliculorum*	NC026102	[40]
		Pyemotoidea	Pyemotidae	*Pyemotes zhonghuajia*	OM396909	[41]
Sarcoptiformes	Desmonomatides	Acaroidea	Acaridae	*Rhizoglyphus robini*	NC038058	[42]
		Analgoidea	Pyroglyphidae	*Dermatophagoides pteronyssinus*	NC012218	[43]
		Sarcoptoidea	Psoroptidae	*Psoroptes cuniculi*	NC024675	[44]
		Histiostomatoidea	Histiostomatidae	*Histiostoma feroniarum*	NC038207	[42]

## Data Availability

The datasets generated for this study can be found in GenBank with the following accession codes: NC039813 and PX252369.

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
