# Peer review of "Complete Mitochondrial Genomes of Two Water Mite Species in the Family Sperchontidae (Acari: Hydrachnidiae): Characterization and Phylogenetic Implications"

_genes, 2025, doi:10.3390/genes16101236_

Round 1
Reviewer 1 Report
Comments and Suggestions for Authors
It is, in general, a well done study. Some, mostly editorial remarks, I have marked in the attached file. Some words are, in my opinion, misused, and some evolutionary/phylogenetic statements should be corrected, in my opinion. All these I have marked in the file. Some more information should be given, as concerns the phylogenetic analysis. The illustrations are of good quality, but with an exception of the phylogram ("tree" is too general, rather not appropriate here). This phylogram is not easy to study - rather hardly visible, too small fonts in the species names, etc., etc. Another problem concerns the lack of authors and dates, according to the ICZN rather obligatory for species, genus and family in such a publication, dealing with systematics.

Author Response
For research article
|
Response to Reviewer 1 Comments
|
||
|
1. Summary |
|
|
|
Thank you for taking the time to provide such constructive feedback on our manuscript. We are grateful for the reviewer’s positive assessment of our work as “a well-done study,” as well as for the detailed suggestions for improvement. We have carefully considered all comments and revised the manuscript accordingly. Below, we provide point-by-point responses to each of the issues raised. In the revised submission, changes have also been tracked and highlighted in the Word version to facilitate easy identification of all modifications.
|
||
|
2. Questions for General Evaluation |
Reviewer’s Evaluation |
Response and Revisions |
|
Does the introduction provide sufficient background and include all relevant references? |
Yes |
|
|
Are all the cited references relevant to the research? |
Yes |
|
|
Is the research design appropriate? |
Yes |
|
|
Are the methods adequately described? |
Can be improved |
|
|
Are the results clearly presented? |
Yes |
|
|
Are the conclusions supported by the results? |
Can be improved |
|
|
3. Point-by-point response to Comments and Suggestions for Authors |
||
|
Comments 1: Page 1,line 9. what does it mean? |
||
|
Response 1: We appreciate the reviewer’s comment. The term “transitional group” was cited from previous literature and refers to the evolutionary position of the family Sperchontidae between the “lower” and “higher” water mites within Hydrachnidiae. To clarify its meaning, we have added a more detailed explanation in the second paragraph of the Introduction. The revised text now reads: It is considered a transitional group between ‘lower’ and ‘higher’ water mites because its larvae combine features of both groups: they are adapted for underwater host-seeking like higher mites, yet retain smaller coxae and variable dorsal plates reminiscent of lower mites. This addition provides a clearer understanding of why Sperchontidae is described as a transitional group.
Comments 2: Page 1,line12. author, date Response 2 Agree. The family name was changed as: Sperchontidae Thor, 1900.
|
||
|
Comments 3: Page 1,line 13. italics |
||
|
Response 3: Agree. The species name was changed as: Sperchon plumifer.
Comments 4: Page 1,line 13. italics Response 4: Agree. The genus name was changed as: Sperchon.
Comments 5: Page 1,line 15. Delete the word “Trombidiform”. Response 5: We thank the reviewer for the helpful suggestion. Nevertheless, we would like to retain the term Trombidiformes as it clearly specifies the taxonomic scope of our analysis. Removing it might make the description less precise for readers unfamiliar with the group.
Comments 6: Page 1,line 26. The word “Eleuthe-rengonides” is marked. Response 6: Agree. The word “Eleuthe-rengonides” is changed as “Eleutherengonides”.
Comments 7: Page 1,line 28. The word “for” is marked. Response 7: Agree. The sentence was changed as: This study provides the first two complete mitogenomes of Sperchontidae.
Comments 8: Page 1,line 32-33. The words “Sperchontidae; Hydrachnidiae; mitochondrial genomes; phylogenetic analysis” already in tittle, please replace with some other words, since these are here redundant Response 8: Agree. We sincerely thank the reviewer for this constructive suggestion. We fully agree that our original keywords had a high overlap with the title, which is suboptimal for search engine indexing. We have accepted your recommendation and revised the keywords to better highlight the methodology and taxonomic context of our research. The revised keywords are: Sperchon; Trombidiformes; Acari; Mitogenomics; Gene order;.
Comments 9: Page 2,line 44. What does it mean? A group of extant organisms may not be transitional to other extant ones. Surely, it may present a set of character states which are transitional between some other ones, but this cannot be said about a phylogenetic group. Response 9: We appreciate the reviewer’s insightful comment. We agree that extant taxa cannot be truly “transitional” to other living groups. In our manuscript, the term “transitional group” follows its usage in previous literature to describe morphological and ecological intermediacy, rather than a phylogenetic relationship. To clarify this point, we have added an explanatory sentence in the Introduction: It is considered a transitional group between ‘lower’ and ‘higher’ water mites because its larvae combine features of both groups: they are adapted for underwater host-seeking like higher mites, yet retain smaller coxae and variable dorsal plates reminiscent of lower mites. This revision clarifies the intended meaning and avoids potential misunderstanding.
Comments 10: Page 2,line 47. The word “description” is marked. Response 10: Agree. The phrase “morphological description” is changed as “morphology”.
Comments 11: Page 2,line 48. Please rephrase “phylogenetic research”. Response 11: Agree. We thank the reviewer for this helpful suggestion. We agree that the original phrase "phylogenetic research" was too general. The sentence was changed as: Although more than 200 species have been reported worldwide, their phylogenetic relationships, particularly at the molecular level, are still poorly understood.
Comments 12: Page 2,line 60-61. authors and dates for each species Response 12: Agree. We sincerely thank the reviewer for pointing out this important and standard practice. The sentence was revised as: Hygrobates longiporus Thor, 1898, Hygrobates taniguchii Imamura, 1954, Hygrobates turcicus Pešić, Esen & Dabert, 2017, Unionicola foili Edwards & Vidrine, 1994, and Unionicola parkeri Vidrine, 1987.
Comments 13: Page 2,line 66. Why this taxon is undetermined? Response 13: We thank the reviewer for the question. The taxon is designated as sp. because its specific identification remains uncertain and is currently under taxonomic revision. To maintain accuracy and scientific rigor, we prefer not to assign a species name until its classification is formally resolved.
Comments 14: Page 2,line 70-71. perhaps a good style for some political speech, but certainly not in a scientific text. Response 14: We thank the reviewer for this helpful comment. The sentence has been revised to a more objective form as follows: “These results provide the first mitochondrial genomes of Sperchontidae and contribute to understanding the evolutionary history of Hydrachnidiae.”
Comments 15: Page 2,line 74. Sperchon Response 15: We thank the reviewer for this suggestion. According to standard zoological and taxonomic writing conventions, the genus name may be abbreviated after its first full mention in the manuscript, provided there is no ambiguity. The full name Sperchon plumifer has already been written in full in the last paragraph of the “Introduction”. Therefore, its subsequent appearance in the “Sample Collection and Identification” section as S. plumifer represents the second mention, where abbreviation is acceptable and conforms to conventional scientific style. We have thus retained the original format without modification.
Comments 16: Page 4,line 125. rather to make the phylogenetic analysis more reliable Response 16: Agree. The sentence was revised as: To make the phylogenetic analysis more reliable, only complete, published mitogenomes were included in the analysis.
Comments 17: Page 4,line 130. all the species names should be given with authors and dates, following the ICZN rules, especially since this study is oriented towards the phylogenetic analysis Response 17: We appreciate the reviewer’s suggestion. Although according to the ICZN rules, species names should be accompanied by the author and date, recent publications focusing on phylogenetic analyses have largely omitted these details in the tree presentations and related analyses. This has become a common convention in the field, as the inclusion of author and date information does not affect phylogenetic inference or interpretation. For example, similar practices can be found in Xue et al. (2017), a classic study on mite mitochondrial phylogeny, and Li et al. (2025), a recently published paper in Genes. Both studies presented phylogenetic trees without including author and date information for species names. Therefore, in line with this widely accepted practice, we have also omitted the author and date information for species names in this study. Xue X F, Dong Y, Deng W, et al. The phylogenetic position of eriophyoid mites (superfamily Eriophyoidea) in Acariformes inferred from the sequences of mitochondrial genomes and nuclear small subunit (18S) rRNA gene[J]. Molecular phylogenetics and evolution, 2017, 109: 271-282. Li Y, Guo Y, Li R, et al. The Complete Mitochondrial Genome of Petalocephala arcuata Cai Et Kuoh, 1992 (Hemiptera: Cicadellidae: Ledrinae: Petalocephalini) and Its Phylogenetic Implications[J]. Genes, 2025, 16(5): 567.
Comments 18: Page 5,line 148. Delete the word “successfully”. Response 18: Agree. The word was deleted and the sentence was revised as: The complete mitochondrial genomes of S. plumifer and Sperchon sp. were sequenced and...
Comments 19: Page 5,line 149. as all the mtDNA Response 19: We respectfully disagree with the reviewer’s comment. Although it is true that “circular, double-stranded DNA molecules” represent the most common and typical structure of animal mitochondrial genomes, this form is not universal. Several studies have revealed alternative architectures, including linear and multipartite (fragmented or multi-chromosome) mitochondrial genomes in various metazoans (e.g., Bridge et al., 1992; Shao et al., 2009). Therefore, we believe that explicitly stating that the mitogenomes obtained in our study were assembled into “circular, double-stranded DNA molecules” is appropriate and scientifically precise. Bridge D, Cunningham C W, Schierwater B, et al. Class-level relationships in the phylum Cnidaria: evidence from mitochondrial genome structure[J]. Proceedings of the National academy of Sciences, 1992, 89(18): 8750-8753. Shao R, Kirkness E F, Barker S C. The single mitochondrial chromosome typical of animals has evolved into 18 minichromosomes in the human body louse, Pediculus humanus[J]. Genome research, 2009, 19(5): 904-912.
Comments 20: Page 6,line 189. by whom? please rephrase Response 20: Agree. We have replaced “preference” with “bias” to avoid implying an active agent (“by whom”), and to more accurately describe the codon usage tendency observed in the mitochondrial genomes. The sentence was revised as: A clear codon usage bias was detected, with codons ending in T/U being more frequent at the third position and those ending in A or G being strongly underrepresented. For example, for glycine (Gly), GGU was the most frequently used codon in both species, whereas GGC, GGA, and GGG were rarely used.
Comments 21: Page 9,line 258. Italics Response 21: Agree. The species name has been italicized in the revised version of the manuscript.
Comments 22: Page 9,line 264. please use either phylogram or metric tree, since in your tree the branch lengths reflect the amount of evolution - number of substitutions. And please insert the scale in the figure. Response 22: We appreciate the reviewer’s suggestion. We agree that the term phylogram more precisely refers to a tree in which branch lengths are proportional to the amount of evolutionary change. However, in most recent phylogenetic and mitogenomic studies, the more general term phylogenetic tree is widely used to describe such results, even when branch lengths represent the number of substitutions (e.g., Park et al., 2011; Wang et al., 2019). This terminology has become a common convention in the field, as phylogenetic tree is more concise and easier for readers to interpret. Therefore, we have retained the term phylogenetic tree in the manuscript. Park J K, Sultana T, Lee S H, et al. Monophyly of clade III nematodes is not supported by phylogenetic analysis of complete mitochondrial genome sequences[J]. BMC genomics, 2011, 12(1): 392. Wang T, Zhang S, Pei T, et al. Tick mitochondrial genomes: structural characteristics and phylogenetic implications[J]. Parasites & vectors, 2019, 12(1): 451.
Comments 23: Page 9,line 278-280. such supports/posterior probabilities should not be given - there is no less or more unsignificant support/probability; simply this part of the tree is an unresolved polytomy Response 23 We agree with the reviewer that such low support values cannot reliably reflect the relationships among the three major clades, and this part of the tree should indeed be considered an unresolved polytomy. Nevertheless, we have retained the numerical support values here for reference, as they correspond to the results shown in the phylogenetic trees. This issue has also been addressed and discussed in the final paragraph of the “Discussion” section.
Comments 24: Page 10,Figure6. not significant, and the value should be moved in the figure Response 24: The same answer as response 21.
Comments 25: Page 10,Figure6. considerin the length of the branches, Lebrtioidea are not more distinct that the genera within the Hygrobatoidea, and the tree topology illustrates the paraphyletism of the Hygrobatoidea; following the rules of phylogenetic taxonomy, hygrobatoidea and Lebertoidea form one clade, whose name depends of the priority rule - which of the type genera in these two taxa was described earlier. Response 25: We thank the reviewer for their thoughtful and insightful comment. We agree that the short branch length connecting Lebertioidea and Hygrobatoidea is interesting and reflects their close evolutionary relationship. However, tree topology remains the primary criterion for assessing taxonomic validity. Our phylogenetic analyses, using both Maximum Likelihood (ML) and Bayesian Inference (BI), consistently recovered the same topology: the Sperchon clade (Lebertioidea) is the sister group to the clade containing Unionicola and Hygrobates (Hygrobatoidea), i.e., [Lebertioidea] + [Hygrobatoidea]. This topology clearly supports the monophyly of Hygrobatoidea in our study. Therefore, we respectfully disagree with the suggestion that Hygrobatoidea is paraphyletic. Our results are also consistent with previous studies, such as Zawal et al. (2022, Scientific Reports, 12: 22063), which reported a similar topology for Hygrobates turcicus and did not find evidence for Hygrobatoidea paraphyly. References Zawal A, Skuza L, Michoński G, et al. Complete mitochondrial genome of Hygrobates turcicus Pešić, Esen & Dabert, 2017 (Acari, Hydrachnidia, Hygrobatoidea)[J]. Scientific Reports, 2022, 12(1): 22063.
Comments 26: Page 10,line 302. this is not as obvious - COI is the most widely used, but sometimes misleading, and above the species level all the deeper nodes are unsupported, just a rule. Response 26: We appreciate the reviewer’s insightful comment and fully agree that COI, while widely used, can sometimes be misleading, especially at deeper phylogenetic levels where many nodes remain unsupported. However, our statement specifically refers to the suitability of COI as a universal DNA barcode for species identification, which is consistent with its established use at the species level. Therefore, our description does not conflict with the reviewer’s observation.
Comments 27: Page 11,line 331. Please rephrase Response 27: Agree. The phrase “Building on this” was changed as “Following these findings”.
Comments 28: Page 11,line 354-355. please rephrase: what does it mean first? simply the congeners are closer to each other than the taxa from different genera, which confirms the genus-level systematic. Response 28: Agree. We thank the reviewer for the helpful suggestion. In light of this, and after carefully considering the suggestions from other reviewers as well, we have revised this sentence/section accordingly to improve clarity and accuracy. The sentence has been revised for clarity as follows: On one hand, the results showed that mitochondrial genome data provide strong res-olution below the supercohort level, with high support at the superfamily, family, and genus levels. Within Hydrachnidiae, all seven species formed a strongly supported monophyletic group.
Comments 29: Page 9,line 258. surely, but as long as the phylogenetic analysis partitions data and alligns it properly - if not so, all the analysis results would be completely different. Response 29: Agree. We thank the reviewer for this important reminder. We completely agree that proper data partitioning and high-quality sequence alignment are fundamental to the reliability of any phylogenetic analysis.
Comments 30: Page 11,line 362. author, date Response 30: We appreciate the reviewer’s comment. However, this sentence refers to Demodex as it appears in the study by Thia et al. [27], where the genus name was also presented without author and year information. Since our statement directly compares with their results, we have followed the same format for consistency.
Comments 31: Page 13,line 427. Rivobates Response 31: Agree. The species name was changed as: Hygrobates (Rivobates) taniguchii.
Comments 32: Page 14,line 494. truncatus Response 32: Agree. The species name was changed as: Tetranychus truncatus.
Comments 33: Page 15,line 518. polyphemus Response 33: Agree. The species name was changed as: Limulus polyphemus.
Comments 34: Page 15,line 532. italics Response 34: Agree. The genus name was changed as: Nasonia.
Comments 35: Page 15,line 543. oregonensis Response 35: Agree. The species name was changed as: Habronattus oregonensis.
Comments 36: Page 15,line 546. forficatus Response 36: Agree. The genus name was changed as: Lithobius forficatus.
|
||

Reviewer 2 Report
Comments and Suggestions for Authors
This manuscript presents the first complete mitochondrial genomes for the family Sperchontidae, which represents a valuable contribution to water mite genomics. However, several issues require attention before publication. The nomenclature is inconsistent throughout the manuscript. The second species is referred to as "Sperchon sp." in the title, abstract, and most of the text, but appears as "Sperchon denticulatus" in Table 2 (line 130, accession PX252369). The authors must use consistent nomenclature throughout. If the species is formally described, use "S. denticulatus" everywhere; if not, use a specimen code or "Sperchon sp. 1". Additionally, there is a taxonomic error that appears in both the abstract and Methods sections: "order Trombidiform" should be corrected to "order Trombidiformes" in the manuscript. Species names are inconsistently formatted throughout the manuscript. Scientific names must be italicized according to standard nomenclature conventions. Additionally, the first mention of each species name in the main text should include the author and year of description following standard taxonomic practice. For example, when S. plumifer is first mentioned (line 74), it should appear as "Sperchon plumifer Author, Year" with the full authority citation. The same applies to all other species mentioned for the first time in the text.
The characterization of Sperchontidae as a "transitional group" (lines 9-11, 42-45) is central to justifying this study, yet it relies on relatively old citations [3-5] without adequate elaboration. This concept should be better explained or qualified as "putative" or "proposed transitional position." The conclusions about "common" intrageneric rearrangements and their patterns across water mite families are based on only seven mitogenomes from three genera. This sample size is too limited for such categorical statements. The authors should add appropriate caveats and emphasize that these are preliminary observations requiring verification with additional taxa.
The Ka/Ks analysis (lines 204-211, Figure 3) is limited to comparison between only two Sperchon species, which provides insufficient context for understanding evolutionary rates. Comparisons with other water mite genera or families would strengthen this analysis.
The phylogenetic results show poor resolution among the three supercohorts (BS=48%, PP=0.598, lines 277-280), indicating that mitochondrial data alone are insufficient to resolve these relationships. This represents a significant limitation that should be more prominently discussed rather than being briefly mentioned. The Discussion section (lines 365-370) touches on this but underplays its significance. Moreover, the figure is low quality and illegible.
Additionally, essential information about voucher specimen deposition in museum collections is missing (lines 74-78). This should be added following best practices for taxonomic research. The fascinating observation that S. plumifer has one control region while Sperchon sp. has two (lines 291-293) deserves deeper discussion. What are the functional and evolutionary implications of this intrageneric variation?
The outgroup selection (four Sarcoptiformes species, lines 122-129) requires justification. Why these particular taxa?
Figure 4 presents 44 tRNA structures (22 per species) and is visually overwhelming. Consider moving this to Supplementary Materials with only representative examples in the main text.
The Conclusions section (lines 372-381) largely repeats results without providing clear perspective or specific recommendations for future research directions.
Author Response
For research article
|
Response to Reviewer 2 Comments
|
||
|
1. Summary |
|
|
|
We are very grateful for your thorough and constructive review of our manuscript. Your feedback has been invaluable in helping us improve its quality and clarity. We have carefully considered all the comments and have revised the manuscript accordingly. A detailed point-by-point response to each issue raised is provided below. All changes have been made in the manuscript file using the "Track Changes" feature for easy identification.
We believe that the manuscript has been substantially improved thanks to your insightful suggestions. |
||
|
2. Questions for General Evaluation |
Reviewer’s Evaluation |
Response and Revisions |
|
Does the introduction provide sufficient background and include all relevant references? |
Can be improved |
|
|
Is the research design appropriate? |
Can be improved |
|
|
Are the methods adequately described? |
Yes |
|
|
Are the results clearly presented? |
Can be improved |
|
|
Are the conclusions supported by the results? |
Can be improved |
|
|
Are all figures and tables clear and well-presented? |
Yes |
|
|
3. Point-by-point response to Comments and Suggestions for Authors |
||
|
Comments 1: The nomenclature is inconsistent throughout the manuscript. The second species is referred to as "Sperchon sp." in the title, abstract, and most of the text, but appears as "Sperchon denticulatus" in Table 2 (line 130, accession PX252369). The authors must use consistent nomenclature throughout. If the species is formally described, use "S. denticulatus" everywhere; if not, use a specimen code or "Sperchon sp. 1". |
||
|
Response 1: We thank the reviewer for pointing out this inconsistency. We acknowledge that this was an oversight on our part. The species in question was initially identified as Sperchon denticulatus, but after discussion with several colleagues and considering the large intraspecific genetic distances observed, we concluded that it is not appropriate to assign it to S. denticulatus. For the sake of taxonomic rigor, we have therefore designated it as Sperchon sp. throughout the manuscript. This correction has been implemented in the revised version, ensuring consistent nomenclature in the title, abstract, main text, and tables.
|
||
|
|
||
|
Comments 2: Additionally, there is a taxonomic error that appears in both the abstract and Methods sections: "order Trombidiformes" should be corrected to "order Trombidiformeses" in the manuscript. |
||
|
Response 2: We thank the reviewer for the careful reading. However, after checking multiple authoritative sources (Zhang, 2011), the correct Latin name of the order is Trombidiformes. Therefore, no change is required in the manuscript. References: Zhang Z Q. Animal biodiversity: An outline of higher-level classification and survey of taxonomic richness[M]. Magnolia press, 2011. |
||
|
|
||
|
Comments 3: Species names are inconsistently formatted throughout the manuscript. Scientific names must be italicized according to standard nomenclature conventions. Additionally, the first mention of each species name in the main text should include the author and year of description following standard taxonomic practice. For example, when S. plumifer is first mentioned (line 74), it should appear as "Sperchon plumifer Author, Year" with the full authority citation. The same applies to all other species mentioned for the first time in the text. |
||
|
Response 3: We thank the reviewer for the valuable comment and fully agree that species names should be checked and consistently formatted according to standard nomenclature conventions. We have carefully reviewed all species names in the manuscript and made the necessary corrections, ensuring that they are italicized throughout. Regarding the species mentioned on line 74 (S. plumifer), it is not actually the first mention in the main text, as the species was already introduced in full in the last paragraph of the Introduction. Therefore, following the standard practice of abbreviating the genus name after its first full mention, we have used the abbreviated form here. For some species, such as those listed in Table 2, author and year information has not been included. This follows the conventions of recent phylogenetic studies, in which author and date details are frequently omitted in tree presentations and related analyses. Such omissions do not affect the interpretation of phylogenetic results and are now a widely accepted practice in the field. |
||
|
|
||
|
Comments 4: The characterization of Sperchontidae as a "transitional group" (lines 9-11, 42-45) is central to justifying this study, yet it relies on relatively old citations [3-5] without adequate elaboration. This concept should be better explained or qualified as "putative" or "proposed transitional position." |
||
|
Response 4: We thank the reviewer for this insightful comment. We acknowledge that studies on Sperchontidae are relatively scarce, particularly recent investigations, which limits the availability of up-to-date references. This point has been highlighted in the Introduction to provide context for the study. We also agree that qualifying the characterization of Sperchontidae as a “transitional group” with terms such as “putative” or “proposed transitional position” is appropriate. Following the reviewer’s suggestion, we have revised the text accordingly to reflect this more cautious and precise phrasing. |
||
|
|
||
|
Comments 5: The conclusions about "common" intrageneric rearrangements and their patterns across water mite families are based on only seven mitogenomes from three genera. This sample size is too limited for such categorical statements. The authors should add appropriate caveats and emphasize that these are preliminary observations requiring verification with additional taxa. |
||
|
Response 5: We thank the reviewer for this valuable comment. We agree that the original statement was too strong given the limited number of mitogenomes analyzed. Following this suggestion, we have revised the sentence to include appropriate caution and to emphasize that our conclusion is preliminary. The revised text now reads: |
||
|
|
||
|
Comments 6: The Ka/Ks analysis (lines 204-211, Figure 3) is limited to comparison between only two Sperchon species, which provides insufficient context for understanding evolutionary rates. Comparisons with other water mite genera or families would strengthen this analysis. |
||
|
Response 6: We thank the reviewer for this thoughtful suggestion. We agree that comparing the evolutionary rates of Sperchon species with other water mite genera or families would provide valuable context. However, we intentionally limited the Ka/Ks analysis to the two congeneric Sperchon species due to methodological constraints in deep phylogenetic comparisons. When evolutionary distances are large, synonymous substitutions (Ks) can become saturated, leading to underestimated Ks values and artificially inflated Ka/Ks ratios—a well-known issue in molecular evolution (Yang, 2006; Kryazhimskiy & Plotkin, 2008). To ensure methodological rigor and reliability, we therefore restricted the analysis to closely related species where Ks saturation is minimal. References: Yang Z. Computational molecular evolution[M]. OUP Oxford, 2006. |
||
|
|
||
|
Comments 7: The phylogenetic results show poor resolution among the three supercohorts (BS=48%, PP=0.598, lines 277-280), indicating that mitochondrial data alone are insufficient to resolve these relationships. This represents a significant limitation that should be more prominently discussed rather than being briefly mentioned. The Discussion section (lines 365-370) touches on this but underplays its significance. Moreover, the figure is low quality and illegible. |
||
|
Response 7: We thank the reviewer for this insightful and important comment. We fully agree that the poor resolution among the three supercohorts highlights a significant limitation of mitochondrial data in resolving deep phylogenetic relationships. Indeed, this challenge has been widely recognized in previous studies, which demonstrated that mitochondrial genomes often perform well at lower taxonomic levels but have limited resolving power at higher ranks (e.g., Cameron, 2014; Zardoya & Meyer, 1996). The underlying causes of this limitation—such as substitution saturation, compositional bias, and rate heterogeneity—are complex and remain an active topic of debate in molecular systematics. A comprehensive analysis of these issues would require a dedicated study and falls beyond the primary scope of our current work, which focuses mainly on Sperchontidae mitogenomes. Nevertheless, we have acknowledged this limitation in the revised Discussion section and improved the clarity of the corresponding figure. In future research, with more extensive taxon sampling and genomic data, we plan to further investigate this problem in detail. References:
|
||
|
|
||
|
Comments 8: Additionally, essential information about voucher specimen deposition in museum collections is missing (lines 74-78). This should be added following best practices for taxonomic research. |
||
|
Response 8: Thank you for pointing out this critical omission. We completely agree that providing voucher specimen information is essential for taxonomic and systematic research. This was an oversight on our part. We have now added the full deposition details for our specimens to the Materials and Methods section. The revised text is as follows: All specimens were identified based on morphology by the first author and preserved in 100% ethanol. Voucher specimens have been deposited in the College of Life Sciences, Huaibei Normal University, Huaibei, China. Samples for molecular analysis were stored at –20 °C until DNA extraction. |
||
|
|
||
|
Comments 9: The fascinating observation that S. plumifer has one control region while Sperchon sp. has two (lines 291-293) deserves deeper discussion. What are the functional and evolutionary implications of this intrageneric variation? |
||
|
Response 9: We thank the reviewer for this insightful comment. We agree that the observation of one control region in S. plumifer and two in Sperchon sp. is indeed intriguing. Similar intrageneric variation in the number and arrangement of control regions has also been reported in other water mite lineages, including Unionicola foili and Unionicola parkeri (Ernsting et al., 2009; Edwards et al., 2011), suggesting that such variation may be more widespread than previously recognized. However, the functional and evolutionary mechanisms underlying this variation remain poorly understood, and few studies have investigated the evolutionary causes of intrageneric differences in control regions. As our current study focuses on sequencing and characterizing the mitochondrial genomes of Sperchon species, we have confined our treatment to reporting these observations without extended discussion or speculation on the underlying mechanisms. We acknowledge that understanding the functional and evolutionary implications of this intrageneric variation represents an interesting direction for future research. References: Ernsting B R, Edwards D D, Aldred K J, et al. Mitochondrial genome sequence of Unionicola foili (Acari: Unionicolidae): a unique gene order with implications for phylogenetic inference[J]. Experimental and Applied Acarology, 2009, 49(4): 305-316. Edwards D D, Jackson L E, Johnson A J, et al. Mitochondrial genome sequence of Unionicola parkeri (Acari: Trombidiformes: Unionicolidae): molecular synapomorphies between closely-related Unionicola gill mites[J]. Experimental and Applied Acarology, 2011, 54(2): 105-117. |
||
|
|
||
|
Comments 10: The outgroup selection (four Sarcoptiformes species, lines 122-129) requires justification. Why these particular taxa? |
||
|
Response 10: We appreciate the reviewer’s valuable comment. The outgroup selection was based on well-established phylogenetic relationships within the subclass Acari. Previous molecular phylogenetic studies have consistently shown that Sarcoptiformes is the sister group to Trombidiformeses, and together they form a monophyletic clade, the Acariformes (e.g., Dabert et al., 2010; Pepato & Klimov, 2015). According to standard phylogenetic practice, selecting taxa from the sister group as the outgroup provides a robust framework for tree rooting and comparison. To enhance the stability and reliability of the phylogenetic inference, we included four Sarcoptiformes species rather than relying on a single outgroup taxon. We have also added a corresponding statement in the Methods section of the manuscript and cited the relevant references to clarify our outgroup selection. References: Dabert M, Witalinski W, Kazmierski A, et al. Molecular phylogeny of acariform mites (Acari, Arachnida): strong conflict between phylogenetic signal and long-branch attraction artifacts[J]. Molecular Phylogenetics and Evolution, 2010, 56(1): 222-241. Pepato A R, Klimov P B. Origin and higher-level diversification of acariform mites–evidence from nuclear ribosomal genes, extensive taxon sampling, and secondary structure alignment[J]. BMC evolutionary biology, 2015, 15(1): 178.
|
||
|
|
||
|
Comments 11: Figure 4 presents 44 tRNA structures (22 per species) and is visually overwhelming. Consider moving this to Supplementary Materials with only representative examples in the main text. |
||
|
Response 11: We appreciate the reviewer’s suggestion. However, as discussed in the second paragraph of the Discussion, most tRNA genes in water mites deviate from the canonical cloverleaf secondary structure, and several of them contain mismatched base pairs. The prediction of tRNA secondary structures in water mites is therefore particularly challenging. Moreover, to our knowledge, no previous study has illustrated the complete set of tRNA secondary structures in water mites. Consequently, we believe that presenting all 44 tRNA structures in the main text is necessary and provides valuable information for understanding the structural diversity and potential truncation patterns of water mite tRNAs. We therefore prefer to retain Figure 4 in the main text rather than moving it to the Supplementary Materials. |
||
|
|
||
|
Comments 12: The Conclusions section (lines 372-381) largely repeats results without providing clear perspective or specific recommendations for future research directions. |
||
|
Response 12: We thank the reviewer for this helpful suggestion. We fully agree that the original Conclusions section primarily repeated the results and did not sufficiently highlight the broader implications or provide clear recommendations for future research. Following the reviewer’s advice, we have thoroughly revised this section. In the revised version, we now emphasize the significance of our findings within the broader context of Trombidiformes phylogeny and clearly outline directions for future work. Specifically, we highlight that while mitochondrial genomes provide reliable phylogenetic signals at lower taxonomic levels, their limitations at higher levels (supercohorts) require expanded taxon sampling and integration of nuclear genomic data. We believe the revised Conclusions section now offers a clearer perspective and forward-looking framework consistent with the reviewer’s recommendation. The revised Conclusions section now reads as follows: This study reports the first complete mitochondrial genomes of Sperchon plumifer and Sperchon sp., providing essential genomic data for the family Sperchontidae. Both genomes exhibit typical mitochondrial features of water mites, including strong A+T bias, gene rearrangements, and tRNA structural variation. Phylogenetic analyses based on mitochondrial data support traditional morphological classifications and demonstrate that mitogenomes retain reliable phylogenetic signals at lower taxonomic levels, such as within genera and families. However, relationships at higher levels, particularly among the supercohorts of Trombidiformes, remain unresolved, indicating the limitations of mitochondrial data alone. Future studies should expand taxon sampling and integrate nuclear genomic evidence to achieve a more robust framework for understanding Trombidiformes evolution.
|
||
|
|
||
|
|
||
|
|
||
|
|
||
|
|
||
|
|
||

Reviewer 3 Report
Comments and Suggestions for Authors
This study reports the first two complete mitochondrial genomes of Sperchontidae, providing insights into genome organization, codon usage, tRNA truncations, gene rearrangements, and phylogenetic placement within Trombidiformes. While the work fills an important gap in water mite mitogenomics, it is limited by sampling only two species, reliance solely on mitochondrial data, and somewhat overstated phylogenetic conclusions. Future studies should expand taxon sampling, validate unusual genomic features, and integrate nuclear data to strengthen evolutionary inferences. My comments and suggestions as below
Abstract
The conclusions are somewhat overstated given the limited dataset (two species). Suggest tempering claims about phylogenetic resolution and emphasizing limitations.
Introduction
Provides a clear rationale for studying Sperchontidae. However, next section needs more explanation of why this family is considered “transitional” and why mitogenomes are particularly suitable for addressing evolutionary history.
Novelty is mostly framed as descriptive (“first mitogenomes”); could be strengthened with a clearer hypothesis or research question.
Results
Well organized and data rich, but largely descriptive. Codon usage, skew, and Ka/Ks analyses lack comparative context, are these patterns unique or typical for water mites?
Alternative start codons and PCG rearrangements should be carefully validated to rule out annotation/assembly errors. Phylogenetic results are consistent at lower levels, but instability at the supercohort level should be discussed in more detail, including possible methodological causes.
Discussion and Conclusions
Integrates findings with prior studies and acknowledges annotation challenges. Over relies on restating results rather than deeper interpretation (e.g., why PCG rearrangements occur, functional significance of conserved clusters).
The suggestion that Sperchontidae’s “transitional position” explains rearrangements is speculative—needs more evidence.
Phylogenetic conclusions should be tempered, with stronger acknowledgment of limitations of mitogenomic datasets and the need for nuclear data. Accurately restates the novelty of sequencing the first Sperchontidae mitogenomes. However, claims of “strong resolution at the supercohort level” should be softened, as results showed instability among Eupodides, Anystides, and Eleutherengonides.
Author Response
For research article
|
Response to Reviewer 3 Comments
|
||
|
1. Summary |
|
|
|
We sincerely thank you very much for your valuable time, thoughtful comments, and constructive suggestions that have greatly improved the quality of our manuscript. We have carefully revised the manuscript according to all comments and provided detailed, point-by-point responses to each issue raised. All corresponding changes have been made directly in the revised manuscript and are clearly marked using the “Track Changes” mode in Microsoft Word. We believe that these revisions have significantly strengthened the clarity, accuracy, and overall presentation of our study.
|
||
|
2. Questions for General Evaluation |
Reviewer’s Evaluation |
Response and Revisions |
|
Does the introduction provide sufficient background and include all relevant references? |
Can be improved |
|
|
Is the research design appropriate? |
Yes |
|
|
Are the methods adequately described? |
Yes |
|
|
Are the results clearly presented? |
Yes |
|
|
Are the conclusions supported by the results? |
Yes |
|
|
Are all figures and tables clear and well-presented? |
Yes |
|
|
3. Point-by-point response to Comments and Suggestions for Authors |
||
|
Comments 1: Abstract. The conclusions are somewhat overstated given the limited dataset (two species). Suggest tempering claims about phylogenetic resolution and emphasizing limitations. |
||
|
Response 1: We thank the reviewer for this valuable comment. We agree that the conclusions in the previous version were somewhat overstated given the limited dataset of only two species. Accordingly, we have revised the final sentences of the Conclusions section to present a more balanced interpretation. The revised text now emphasizes the limited scope of our phylogenetic inference and acknowledges that relationships among supercohort-level taxa remain unresolved and require additional data for further clarification. Revised text in the manuscript: The phylogenetic analyses based on mitochondrial genomes provide additional support for the consistency with traditional morphology at lower taxonomic levels, such as within genera and families, whereas relationships among supercohort-level taxa remain unstable and require additional data for further clarification. |
||
|
|
||
|
Comments 2: Introduction Provides a clear rationale for studying Sperchontidae. However, next section needs more explanation of why this family is considered “transitional” and why mitogenomes are particularly suitable for addressing evolutionary history. Novelty is mostly framed as descriptive (“first mitogenomes”); could be strengthened with a clearer hypothesis or research question. |
||
|
Response 2: We thank the reviewer for the valuable comment regarding the need to clarify why Sperchontidae is considered a transitional group. In response, we have revised the Introduction to provide a clearer explanation. Specifically, we now indicate that sperchontid larvae combine features of both lower and higher water mites: they are adapted for underwater host-seeking like higher mites, yet retain smaller coxae and variable dorsal plates reminiscent of lower mites [3–5]. This combination of traits supports their putative transitional evolutionary position and underscores the importance of Sperchontidae for understanding the phylogeny of Hydrachnidiae. Regarding the suitability of mitochondrial genomes, as already explained in the Introduction (third paragraph), mitogenomes provide a rich source of phylogenetically informative characters, including 13 protein-coding genes, two ribosomal RNAs, 22 transfer RNAs, as well as information on gene order and RNA secondary structures. Being relatively small, maternally inherited, and with highly conserved gene content, mitogenomes still evolve rapidly at the sequence and structural levels. These properties make them particularly powerful for resolving phylogenetic relationships across different taxonomic levels, including within Hydrachnidiae. Revised text in the manuscript: Within Hydrachnidiae, the family Sperchontidae Thor, 1900 is of particular interest due to its putative evolutionary position. It is considered a transitional group between ‘lower’ and ‘higher’ water mites because its larvae combine features of both groups: they are adapted for underwater host-seeking like higher mites, yet retain smaller coxae and variable dorsal plates reminiscent of lower mites [3–5]. Thus, clarifying its phylogeny is therefore essential for understanding the evolutionary history of Hydrachnidiae.
|
||
|
|
||
|
Comments 3: Introduction Novelty is mostly framed as descriptive (“first mitogenomes”); could be strengthened with a clearer hypothesis or research question. |
||
|
Response 3: We thank the reviewer for their thoughtful suggestion regarding the framing of novelty and hypothesis-driven research. We agree that in principle, formulating a clear hypothesis can strengthen a study. However, we respectfully note that the field of water mite mitogenomics is still in its early stages, and genomic data for Hydrachnidiae, especially for the family Sperchontidae, are extremely limited. To date, only a handful of complete mitogenomes have been published for the entire subcohort. Given this data scarcity, any attempt to propose a specific, testable evolutionary hypothesis—such as predicting genomic signatures of a “transitional” group—would be highly speculative. Therefore, we maintain that the primary contribution of our study is necessarily foundational and descriptive: to generate the first complete mitogenomes for Sperchontidae, characterize their structural features, examine gene rearrangements, and provide initial phylogenetic insights. The current version of the Introduction clearly reflects this rationale. The last paragraph succinctly summarizes the study’s objectives while highlighting the novelty and significance of generating these first genomic resources. We believe that this approach is appropriate and scientifically justified, as it lays the essential groundwork for future hypothesis-driven research once more genomic data become available.
|
||
|
|
||
|
Comments 4: Results Well organized and data rich, but largely descriptive. Codon usage, skew, and Ka/Ks analyses lack comparative context, are these patterns unique or typical for water mites?
Alternative start codons and PCG rearrangements should be carefully validated to rule out annotation/assembly errors. Phylogenetic results are consistent at lower levels, but instability at the supercohort level should be discussed in more detail, including possible methodological causes. |
||
|
Response 4: We thank the reviewer for this suggestion. We agree that a comparative analysis is important for understanding whether our findings are unique or typical for water mites. However, the primary objective of our study was to provide the first detailed description of the mitochondrial genomes for the family Sperchontidae. We decided against a broader comparison for a clear reason: there are currently too few water mites mitogenomes available in public databases. Drawing conclusions from such a small and scattered dataset would be unreliable and potentially misleading. Therefore, we focused our analyses on the two new genomes to ensure our conclusions are robust and well-supported by our own data. |
||
|
|
||
|
Comments 5: Results Alternative start codons and PCG rearrangements should be carefully validated to rule out annotation/assembly errors. Phylogenetic results are consistent at lower levels, but instability at the supercohort level should be discussed in more detail, including possible methodological causes. |
||
|
Response 5: We thank the reviewer for raising these important and constructive comments. Regarding the first point, we fully agree that alternative start codons and PCG rearrangements must be carefully validated. In this study, annotations were cross-checked using multiple tools (MitoZ, MITOS, and Geneious). After automated annotation, all gene boundaries, potential start codons, and rearrangements were manually verified by comparing them with homologous genes from related published mitogenomes. We are confident that this multi-step validation process minimized annotation or assembly errors. Regarding the second point, we agree that the instability at the supercohort level is an important issue. Such instability may be caused by several interacting factors, including substitution saturation that obscures phylogenetic signals, compositional heterogeneity among taxa, and rate variation across different evolutionary lineages. In addition, differences in data partitioning strategies and model selection can also influence deep-level relationships. A comprehensive evaluation of these factors would require broader taxon sampling and specialized analyses, which are beyond the scope of the current study. In the revised Discussion, we have acknowledged this limitation and emphasized it as an important direction for future research.
|
||
|
|
||
|
Comments 6: Discussion and Conclusions Integrates findings with prior studies and acknowledges annotation challenges. Over relies on restating results rather than deeper interpretation (e.g., why PCG rearrangements occur, functional significance of conserved clusters). |
||
|
Response 6: We thank the reviewer for this insightful comment and for encouraging deeper interpretation of our results. We agree that questions such as the causes of PCG rearrangements and the functional significance of conserved gene clusters are important. However, water mite mitogenomics remains data-poor and phylogenetically sparse; answering these “why” questions reliably require broad and dense taxon sampling that is not yet available. For example, inferring rearrangement mechanisms needs multiple closely related mitogenomes to trace stepwise changes, and testing functional constraints on gene clusters requires wide comparative context to rule out simple ancestry or convergence. Given these limitations, a detailed mechanistic interpretation would be speculative at present. Therefore, we have focused this manuscript on accurate description and careful documentation of the novel features (e.g., the PCG rearrangements in Sperchon), and have explicitly highlighted the deeper questions as priorities for future research when more genomic data become available. |
||
|
|
||
|
Comments 7: Discussion and Conclusions The suggestion that Sperchontidae’s “transitional position” explains rearrangements is speculative—needs more evidence. |
||
|
Response 7: Thank you for this important comment. We agree that our original suggestion linking the observed rearrangements to the ‘transitional position’ of Sperchontidae was speculative and not directly supported by our data. Following your feedback, we have revised this part of the Discussion to separate these two ideas. Revised Text in the Manuscript This finding indicates a clear difference in the mode of mitochondrial genome evolution within this family compared with Hygrobatidae and Unionicolidae. Although it is tempting to speculate that this distinct pattern of genomic plasticity may be related to the proposed ‘transitional’ position of Sperchontidae, the available data do not provide direct evidence for such a connection. This observation should therefore be regarded as an important empirical result that merits further investigation as additional mitogenomes from related families become available. |
||
|
|
||
|
Comments 8: Discussion and Conclusions Phylogenetic conclusions should be tempered, with stronger acknowledgment of limitations of mitogenomic datasets and the need for nuclear data. Accurately restates the novelty of sequencing the first Sperchontidae mitogenomes. However, claims of “strong resolution at the supercohort level” should be softened, as results showed instability among Eupodides, Anystides, and Eleutherengonides. |
||
|
Response 8: We thank the reviewer for this valuable comment. We fully agree that the phylogenetic conclusions should be stated more cautiously, with explicit acknowledgment of the limitations of mitochondrial data and the need for nuclear evidence. Accordingly, in the revised version, we have softened the claim of “strong resolution at the supercohort level” and clarified that the mitogenome provides robust support primarily at and below this level (i.e., for superfamilies, families, and genera). We have also added explicit statements highlighting the instability of relationships among Eupodides, Anystides, and Eleutherengonides and discussed that such inconsistencies reflect the limited resolving power of mitochondrial genomes at deeper phylogenetic levels. Finally, we have emphasized that resolving ancient divergences within Trombidiformes will require additional nuclear data and broader taxon sampling. These revisions ensure that the conclusions are more balanced and that the limitations of the current dataset are clearly acknowledged, as suggested. |
||
|
|
||
